# Federated Graph Learning for Cross-Domain Recommendation

**Ziqi Yang**[1,2]**, Zhaopeng Peng**[1,2]**, Zihui Wang**[1,2]**, Jianzhong Qi**[3]**, Chaochao Chen**[4]**,**
**Weike Pan**[5]**, Chenglu Wen**[1,2]**, Cheng Wang**[1,2]**, Xiaoliang Fan**[1,2]*

[1]Fujian Key Laboratory of Sensing and Computing for Smart Cities, Xiamen University, China
[2]Key Laboratory of Multimedia Trusted Perception and Efficient Computing,
Ministry of Education of China, Xiamen University, China
[3]School of Computing and Information Systems, The University of Melbourne, Australia
[4]College of Computer Science and Technology, Zhejiang University Hangzhou, China
[5]College of Computer Science and Software Engineering, Shenzhen University Shenzhen, China
`{yangziqi,pengzhaopeng,wangziwei}@stu.xmu.edu.cn`
`{clwen,cwang,fanxiaoliang}@xmu.edu.cn`
`jianzhong.qi@unimelb.edu.au, zjuccc@zju.edu.cn, panweike@szu.edu.cn`

## Abstract

Cross-domain recommendation (CDR) offers a promising solution to the data sparsity problem by enabling knowledge transfer between source and target domains. However, many recent CDR models overlook crucial issues such as privacy as well as the risk of negative transfer (which negatively impact model performance), especially in multi-domain settings. To address these challenges, we propose FedGCDR, a novel federated graph learning framework that securely and effectively leverages positive knowledge from multiple source domains. First, we design a positive knowledge transfer module that ensures privacy during inter-domain knowledge transmission. This module employs differential privacy-based knowledge extraction combined with a feature mapping mechanism, transforming source domain embeddings from federated graph attention networks into reliable domain knowledge. Second, we design a knowledge activation module to filter out potential harmful or conflicting knowledge from source domains, addressing the issues of negative transfer. This module enhances target domain training by expanding the graph of the target domain to generate reliable domain attentions and fine-tunes the target model for improved negative knowledge filtering and more accurate predictions. We conduct extensive experiments on 16 popular domains of the Amazon dataset, demonstrating that FedGCDR significantly outperforms state-of-the-art methods. We open source the code at `https://github.com/LafinHana/FedGCDR`.

## 1 Introduction

Cross-domain recommendation (CDR) has emerged as an effective solution for mitigating data sparsity in recommender systems [1; 2; 3; 4; 5]. CDR operates by integrating auxiliary information from source domains, thereby enhancing recommendation relevance in the target domain. Recently, to address data privacy constraints, many privacy-preserving CDR frameworks have been proposed [6; 7; 8; 9], which achieve strong performance under the assumptions of **data sparsity and a dual-domain model (i.e., typically involving a single source domain and a single target domain)**.

In this paper, we focus on a more generic scenario of **Broader-Source Cross-Domain Recommendation (BS-CDR)**, which integrates knowledge from more than two source domains while

---

*The corresponding author.

38th Conference on Neural Information Processing Systems (NeurIPS 2024).

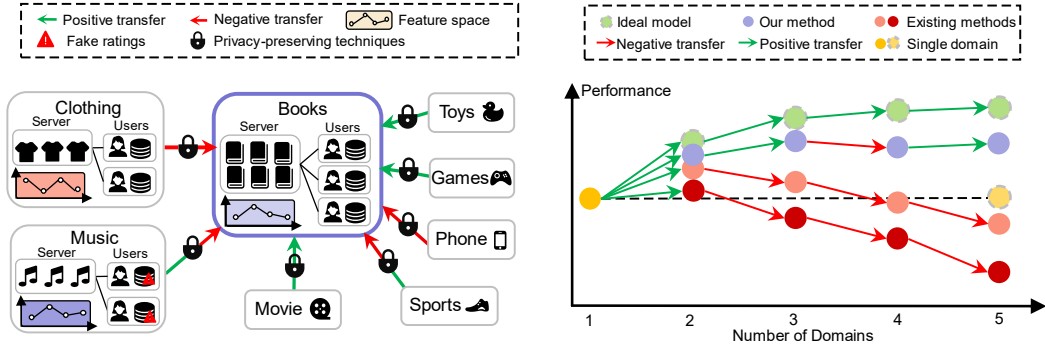

|  |  |
|---|---|
| (a) The BS-CDR scenario. | (b) The performance affected by the number of domains |

Figure 1: (a) In order to obtain accurate recommendations in the Books domain, we aim to exploit user preferences (i.e., knowledge of external domains should be fully utilized, e.g. Movie, Toys, and Games domains). However, with the influence of lossy privacy-preserving techniques, the results of the transfer could be negative (e.g., the Music domain with low-quality data). (b) There is a diminishing marginal effect on the growth rate of the model performance with pure positive knowledge, while NT accumulates with an increasing number of source domains. Consequently, the performance of existing methods declines and is worse than that of a single domain model.

preserving privacy. Given the diverse nature of user preferences, it is essential to gain a more holistic understanding of user interests by incorporating user behaviors from diversified domains [10; 11]. For example, in Figure 1a, a user who enjoys certain types of books might also enjoy movies, toys, and games in similar genres. However, incorporating more domains while preserving privacy poses challenges to counteract negative transfer (NT), which is a phenomenon of transferring knowledge from a source domain that negatively impacts the performance of the recommender model in the target domain [12]. Suppose that the Books domain in Figure 1a is the target domain. The Clothing domain causes NT because of the domain discrepancy. While the Music domain is supposed to transfer positive knowledge, it might also lead to NT because of lossy privacy-preserving techniques applied to broader source domains. As a result, the influx of negative knowledge accumulated from the source domains will poison the model performance of the target domain in BS-CDR scenarios.

To mitigate the NT issue, attention mechanisms have been widely leveraged, either in an explicit (e.g., determine domain attentions by predefined domain features [13; 14]) or implicit manner (e.g., employ hyper-parameters [7; 15]). Several other studies [16; 17; 18], ensure positive transfer by passing only domain-shared features. However, existing methods cannot be directly applied to BS-CDR due to two major challenges. **First**, inadequate privacy preservation (**CH1**). Both intra-domain and inter-domain privacy must be carefully considered in BS-CDR. As depicted in Figure 1a, BS-CDR relies on extensive knowledge transfer, risking simultaneous privacy leakages across broader source domains (inter-domain privacy) [9; 19; 20; 21]. Additionally, concerns over centralized data storage may prevent users from sharing sensitive rating data (intra-domain privacy). **Second**, accumulative negative transfer (**CH2**). Adjusting attention-related hyper-parameters for a large number of source domains in BS-CDR scenarios is extremely difficult, as well as predefined or domain-shared features cannot accommodate complex domain diversities. In addition, the use of various lossy privacy-preserving techniques can further degrade the quality of the transferred knowledge, complicating the achievement of positive transfer. Consequently, the impact of NT can inevitably intensify with an increasing number of source domains [2] and the performance of CDR models can decline to levels lower than those of single-domain models, as shown in Figure 1b.

To address the challenges of privacy (CH1) and NT (CH2) in BS-CDR, we propose Federated Graph learning for Cross-Domain Recommendation (FedGCDR). It follows a **horizontal-vertical-horizontal pipeline** [6] and consists of **two key modules**. First, the positive knowledge transfer module aims to safeguard inter-domain privacy and mitigate potential NT before transfer. This module adopts differential privacy (DP) [22] with a theoretical guarantee and aligns the feature spaces to facilitate positive knowledge transfer. Second, the positive knowledge activation module is engaged to further alleviate NT. Specifically, it expands the local graph of the target domain by incorporating virtual social links, enabling the generation of domain attentions. Additionally, it

performs target model fine-tuning to optimize the broader-source CDR. Extensive experiments on 16 popular domains of the Amazon benchmarks demonstrate that FedGCDR outperforms all baseline methods in terms of recommendation accuracy.

Our contributions are summarized as follows:

- We introduce FedGCDR, a novel federated graph learning framework for CDR that provides high-quality BS-CDR recommendations while safeguarding both user privacy and domain confidentiality;

- We propose two key model, i.e., the positive knowledge transfer module and the positive knowledge activation module. The first transfer module ensure privacy and positive knowledge flows via privacy-preserving knowledge extraction and feature mapping. The second activation module filter harmful information via graph expansion, target domain training and target model fine-tuning;

- We conduct extensive experiments on 16 domains of the Amazon datasets that confirm the effectiveness of FedGCDR in terms of recommendation accuracy.

## 2 Related work

### 2.1 Cross-domain recommendation

CDR utilizes auxiliary information from external domains to alleviate the data sparsity problem and effectively improve recommendation quality. Li et al. [23] enrich domain knowledge by transferring user-item rating patterns from source domains to target domains. Man et al. [15] and Elkahky et al. [24] augment entities' embeddings in the target domain by employing a linear or multilayer perceptron (MLP)-based nonlinear mapping function across domains. Liu et al. [25] address the review-based non-overlapped recommendation problem by attribution alignment. Zhao et al. [18] improve the recommendation quality of multi-sparse-domains by mining domain-invariant preferences. Liu et al. [26] achieve knowledge transfer without overlapping users by mining joint preferences. Chen et al. [19] and Liu et al. [27] avoid intermediate result privacy leakage during cross-domain knowledge transfer by employing DP. In these works, the NT problem is often ignored because most of them assume a carefully selected dual-domain scenarios or limited multi-domain scenarios where NT is not evident. We aim to solve the NT problem in complex BS-CDR scenarios.

### 2.2 Federated recommendation

Recently, federated learning (FL) [28; 29; 30; 31; 32] has been widely adopted to tackle the privacy issue in recommender system. Chai et al. [33] adopt FL to classic matrix factorization algorithm and utilize homomorphic encryption to avoid the potential threat of privacy disclosure. Later, Wu et al. [34] explore the application of federated graph neural networks (GNN) models to improve the recommendation quality and ensure user privacy. To utilize sensitive social information, Liu et al. [8] adopt local differential privacy (LDP) and negative sampling. More recent studies use vertical federated learning (VFL) to protect company's privacy in recommender system. Mai et al. [35] utilize random projection and ternary quantization to ensure privacy preservation in VFL. In CDR, Chen et al. [9] design a dual-target VFL CDR model with orthogonal mapping matrix and LDP for organizations' privacy preservation. Liu et al. [36] design a graph convolutional networks (GCN)-based federated framework to learn user preference distributions for more accurate recommendations. To ensure user privacy in CDR, Liu et al. [6] utilize a VAE-based federated model to mine user preference with data stored locally. Wu et al. [7] design a personal module and a transfer module to provide personalized recommendation while preserving user privacy. These existing works, especially federated CDR frameworks, consider only one type of privacy (intra- or inter-domain). We aim to provide both intra-domain and inter-domain privacy.

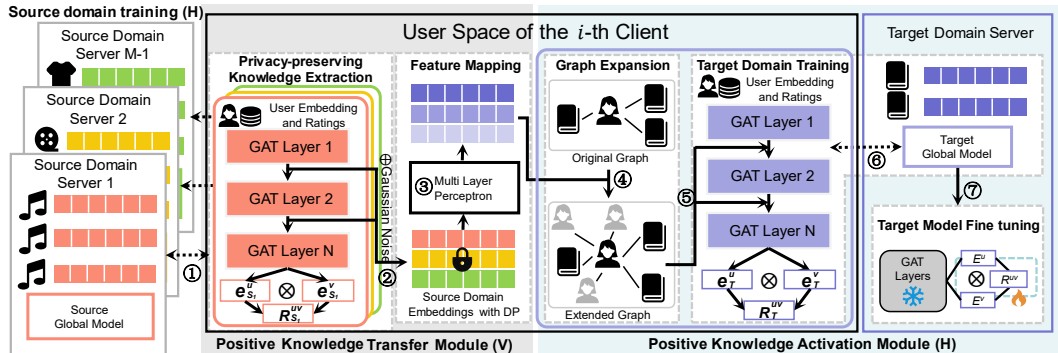

Figure 2: An overview of FedGCDR. It consists of two key modules and follows a HVH pipeline: (1) Source Domain Training (Horizontal FL): ① Each source domain maintains its graph attention network (GAT)-based federated model. (2) Positive Knowledge Transfer Module (Vertical FL): ② Source domain embeddings are extracted from GAT layers and perturbed with Gaussian noise. ③ The multilayer perceptron aligns the feature space of source domain embeddings and target domain embeddings. (3) Positive Knowledge Activation Module (Horizontal FL): ④ Local graph is expanded with source domain embeddings. ⑤ Enhanced federated training of the target domain is achieved through the expanded graph. ⑥ The target domain maintains its GAT-based federated model. ⑦ The target domain freezes the GAT layer and fine tunes the model.

## 3 Methodology

### 3.1 Problem definition

We consider $M$ ($M>3$) domains participating in the CDR process. The domains are divided into M-1 source domains $\mathcal{D}^{S_1}, \mathcal{D}^{S_2}, ..., \mathcal{D}^{S_{M-1}}$ and one target domain $\mathcal{D}^T$. Each domain is assigned a domain server to conduct intra-domain model training. $\mathcal{U}$ is the user set across all the domains, $\mathcal{U} = \mathcal{U}_1 \bigcup \mathcal{U}_2 \bigcup ... \bigcup \mathcal{U}_M$, where $\mathcal{U}_i$ denotes the user set of domain $i$. We assume that users partially overlap between domains. Each user is treated as an individual client. User space refers to the virtual space in the user's device containing domain models distributed from each domain server. Meanwhile, $\mathcal{V}_i$ is the item set of domain $i$. Let $\mathbf{R}^i \in \mathbb{R}^{|\mathcal{U}_i| \times |\mathcal{V}_i|}$ be the observed rating matrix of the $i$-th domain. We consider top-K recommendation, i.e., we learn a function to estimate the scores of unobserved entries in the rating matrix, which are later used for item ranking and recommendations. Our goal is to achieve highly accurate recommendations in the target domain.

### 3.2 Framework of FedGCDR

#### 3.2.1 Overview

The overall framework of FedGCDR is shown in Figure 2. FedGCDR follows a Horizontal-Vertical-Horizontal (HVH) pipeline, and its two horizontal FL stages ensure the intra-domain privacy. Our two key modules focus on the vertical stage and the second horizontal stage: (1) The positive knowledge transfer module preserves the inter-domain privacy by DP and alleviates NT by feature mapping. (2) The positive knowledge activation module filters out potential harmful or conflicting knowledge from the source domains. Specifically, we expand the local graph of the target domain by virtual social links, such that the target domain graph attention network (GAT) model could generate reliable domain attention based on the expanded graph. After target domain GAT model training, we further mitigate NT by adopting a fine-tuning stage.

**Horizontal-Vertical-Horizontal pipeline** The HVH pipeline contains three stages with switching federated settings. The first horizontal stage refers to the source domain training in which source domain servers individually interact with its domain users (clients). The private rating information is stored within each client, while the clients exchange model and gradients to train a domain-specific global model. The next two stages correspond to our two key modules (vertical positive knowledge transfer module and horizontal positive knowledge activation module), which we will cover in detail in the following subsections. It's important to note that the vertical positive knowledge transfer

module is completely computed in each client's user space (their personal devices), thus reducing communication overheads. This is because the needed source domain knowledge can be extracted from local source models on each client which are distributed during the first horizontal source domain training stage.

Following the HVH pipeline, we achieve: (1) Privacy enhancement. The two horizontal stages can provide intra-domain privacy preservation, while we further ensure inter-domain privacy by applying DP to the vertical stage. In the mean time, servers are not involved in the knowledge transfer process (i.e., the positive knowledge transfer module), making them unaware of user interactions in other source domains. (2) Communication efficiency. Cross-domain knowledge transfer does not require additional communication overhead.

**Intra-domain GAT-based federated model** We adopt a GAT-based [37; 38; 39] federated framework as the underlying model for our intra-domain recommender system. The horizontal paradigm avoids centralized storage of user ratings to ensure intra-domain privacy (**CH1**). In the initial step, each user and item is offered an ID embedding of size $d$, denoted by $\mathbf{e}_u^0, \mathbf{e}_v^0 \in \mathbb{R}^d$ respectively. The embedding is passed through $L$ message propagation layers [40; 41; 42]. For the $l$-th layer:

$$\mathbf{e}_u^{l+1} = \sigma(\mathbf{W}^l(a_{uu}^l \mathbf{e}_u^l + \sum_{v \in N_u} a_{uv}^l \mathbf{e}_v^l)), \tag{1}$$

where $N_u$ is the neighbor set of $u$, $\mathbf{W}^l$ is a learnable weight matrix, $a_{uu}^l$ and $a_{uv}^l$ are the importance coefficients computed by the attention mechanism:

$$a_{uv}^l = \frac{exp(LeakyReLU(\alpha(\mathbf{W}\mathbf{e}_u^l||\mathbf{W}\mathbf{e}_v^l))}{\sum_{v' \in N_u \bigcup u} exp(LeakyReLU(\alpha[\mathbf{W}\mathbf{e}_u^l||\mathbf{W}\mathbf{e}_{v'}^l])}, \tag{2}$$

where $\alpha$ is the weight vector. Inspired by LightGCN [43], we discard feature transformation and nonlinear activation for better model efficiency and learning effectiveness:

$$e_u^{l+1} = a_{uu}^l \mathbf{e}_u^l + \sum_{v \in N_u} a_{uv}^l \mathbf{e}_v^l, \tag{3}$$

$$a_{uv} = \frac{exp(\alpha(\mathbf{e}_u^l||\mathbf{e}_v^l))}{\sum_{v' \in N_u \bigcup u} exp(\alpha(\mathbf{e}_u^l||\mathbf{e}_{v'}^l))}. \tag{4}$$

In each source domain, the domain server and corresponding users collaboratively train a GAT-based federated model. The training process follows the horizontal federated learning (HFL) paradigm in which only the model and gradients are exchanged considering intra-domain privacy. We will not detail the horizontal federation model (e.g., further privacy guarantee and more high-order information) as it is a well established FL model and not our novel contribution. This model can be replaced by other GAT-based FL models [34; 44] as well.

### 3.2.2 Positive knowledge transfer module

After the source domain training, we obtain a series of source models in individual client's user space. Our positive knowledge transfer module then prepares positive knowledge to be transferred from each source domains $\mathcal{D}^S$ to the target domain $\mathcal{D}^T$, while protecting inter-domain privacy (**CH1**). Specifically, suppose an individual user (client) $u$ and a source domain $\mathcal{D}^{S_i}$, we transfer the user $u$'s embedding matrix $\mathbf{X}_{S_i} \in \mathbb{R}^{L \times d}$. Take the row $l$ of the matrix (i.e., $\mathbf{x}_{S_i}^l$) as an example, it is the user $u$'s embedding output by the $l$-th message propagation layer ($e_u^l$). In an ideal scenario (i.e., we transfer totally positive knowledge without taking inter-domain privacy into account) [6], embedding matrices from different source domains can be directly used to enhance target domain local training in client $u$. By utilizing the source domain embeddings, $u$'s final target domain embedding $\mathbf{e}_T^l$ of layer $l$ is:

$$\mathbf{e}_T^l = f_T(\mathbf{x}_T^l, \mathbf{x}_{S_1}^l, ..., \mathbf{x}_{S_{M-1}}^l), \quad l \in [1, L] \tag{5}$$

where $f_T(\cdot)$ is the function that the target domain aggregates the knowledge of the source domains and we will give its final expression in Subsection 3.2.3. In this process, the transfer of knowledge between domains takes place entirely in the user $u$'s local space. Such a fully localized mode of knowledge transfer avoids additional communication overhead and potential privacy issues [6]. However, this direct embeddings transfer does not meet the privacy and NT constraints in BS-CDR scenarios.

**Privacy-preserving knowledge extraction** In existing CDR frameworks, the user or item embedding was shared as knowledge [9; 15; 6], which neglects inter-domain privacy. In a GNN-based approach, such direct transfers are subject to privacy attacks. Each message propagation layer can be viewed as a function with user and item embeddings as input. An attacker can easily obtain the user's private rating matrix based on these embeddings. We apply DP to the source domain embeddings $\mathbf{x}_{S_i}$ [22; 45] to safeguard inter-domain privacy.

THEOREM 1. *By introducing Gaussian noise into the source domain embeddings, the reconstructed data from the ideal attack deviates from the actual data, therefore preventing a perfect reconstruction.*

In FedGCDR, we incorporate the Gaussian mechanism with the source domain embeddings $\mathbf{x}_{S_i}$ to obtain $\hat{\mathbf{x}}_{S_i}$ for knowledge transfer. Detailed privacy analysis is included in Appendix A.

**Feature mapping** User features could represent personal preferences and are influenced by domain features. The discrepancy of domains leads to the heterogeneity of feature space between domains which means that source domain embeddings cannot be utilized directly by the target domain. Man et al. [15] show that there exists an underlying mapping relationship between the latent user matrix of different domains, which can be captured by a mapping function. In order to alleviate NT, we adopt a series of MLP to explore mapping functions for each source domain. Adding Gaussian noise and feature mapping, Equation (5) becomes:

$$\mathbf{e}_T^l = f_T(\mathbf{x}_T^l, MLP_1(\hat{\mathbf{x}}_{S_1}^l), ..., MLP_{M-1}(\hat{\mathbf{x}}_{S_{M-1}}^l)). \tag{6}$$

To learn more effective mapping function, we adopt a mapping loss term:

$$l_m = \sum_{i=1}^{M-1} \sum_{l=1}^{L} ||\mathbf{x}_T^l - MLP_i(\hat{\mathbf{x}}_{S_i}^l)||^2, \tag{7}$$

### 3.2.3 Positive knowledge activation module

After the aforementioned operations, the target domain obtains a list of source domain matrices $\hat{\mathbf{X}}_{S_1}, \hat{\mathbf{X}}_{S_2}, ..., \hat{\mathbf{X}}_{S_{M-1}}$. The rows of the matrices represent $MLP_i(\hat{\mathbf{x}}_{S_i}^l)$. It is worth noting that for source domains where a user has no rating, $\hat{\mathbf{X}}_{S_i}$ is a Gaussian noise matrix and our motivation is: (1) no rating may also suggest a preference; (2) this is beneficial for enhancing the model's capability to filter noise and identify NT. With knowledge from the source domains, the purpose of the positive knowledge activation module is to alleviate NT following the knowledge transfer. (**CH2**). Although we have aligned the feature space in the previous module, the Gaussian noise that has been fed to the target domain with source domain embedding matrices leads to potential NT. How to utilize the transferred knowledge remains a great challenge.

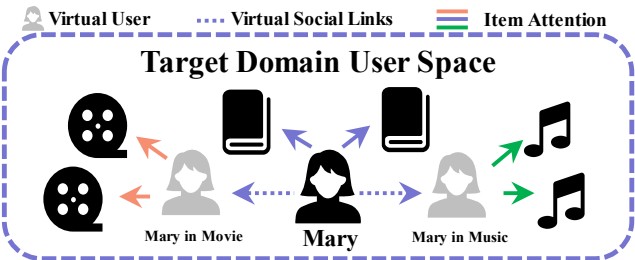

Figure 3: Illustration of target domain graph expansion. The virtual users are constructed with the source domain embeddings from the Movie domain and the Music domain. The attentions generated by social links to the virtual user can be regarded as the domain attentions.

**Graph expansion and target domain training** To alleviate NT, common approaches are to generate domain attention by predefined domain features [13; 46; 47] or to control the transfer ratio of source domains by Writefull [7]. These methods are only applicable to a limited number of domains and have excessive human intervention. In FedGCDR, we take an attention-based approach. First, we expand $u$'s (Mary's) local graph of the target domain as shown in Figure 3.

For the source domain embedding matrices $\hat{\mathbf{X}}_{S_1}, \hat{\mathbf{X}}_{S_2}, ..., \hat{\mathbf{X}}_{S_{M-1}}$, we represent them as $M-1$ virtual users. Since the virtual users constructed from source domain embeddings represent the same individual $u$, they share correlated preferences, with their features (i.e., embeddings) characterizing $u$'s preferences. Inspired by social recommendation [48; 49; 50], we consider that there is an implicit social relationship between virtual users and the actual user $u$, because of the correlation in their preferences. Then, we build virtual social links between them to expand the original target domain graph. Second, by incorporating this expanded graph into target domain training, the GAT model generates corresponding attention coefficients for the virtual users, which can be interpreted as domain-specific attentions. Leveraging the domain attention coefficients, the target domain can focus on domains that transfer positive knowledge and we can finally give $f_T(\cdot)$:

$$f_T(x_T^l, MLP_1(\hat{\mathbf{x}}_{S_1}^l), ..., MLP_{M-1}(\hat{\mathbf{x}}_{S_{M-1}}^l)) = a_{uu}^l \mathbf{x}_T^l + \sum_{v \in N_u} a_{uv}^l \mathbf{e}_v^l + \sum_{i=1}^{M-1} a_i^l MLP_i(\hat{\mathbf{x}}_{S_i}^l), \quad (8)$$

where $a_i^l$ is the domain attention of source domain $i$ generated by the $l$-th layer. Beside, we introduce a social regularization term to strengthen the virtual social links:

$$l_s = \sum_{l=1}^{L} \| \mathbf{x}_T^l - \frac{\sum_{i=1}^{M-1} Sim(\mathbf{x}_T^l, \hat{\mathbf{x}}_{S_i}^l) \times \hat{\mathbf{x}}_{S_i}^l}{\sum_{i=1}^{M-1} Sim(\mathbf{x}_T^l, \hat{\mathbf{x}}_{S_i}^l)} \|^2, \quad (9)$$

the function $Sim(\cdot)$ calculates the cosine similarity [48].

Through the graph expansion, we achieve: (1) dynamic domain attentions that focus on positive source domain knowledge to alleviate NT; (2) attention generation by GAT, eliminating the need for additional interventions such as hyper-parameter tuning or feature engineering.

For top-$k$ recommendation, we adopt a widely-used inner product model to estimate the value of target domain rating $\mathbf{R}_{uv}^T$, which is the interaction probability between a pair of user $u$ and item $v$:

$$\hat{\mathbf{R}}_{uv}^T = Sigmoid(\mathbf{e}_T^u \cdot \mathbf{e}_T^v), \quad (10)$$

where $\mathbf{e}_T^u$ and $\mathbf{e}_T^v$ are the final user and item embeddings output by GAT. Our objective function consists of three terms as follows:

$$L_{GAT} = BCELoss(\hat{\mathbf{R}}_{uv}^T, \mathbf{R}_{uv}^T) + \frac{\alpha}{2} l_m + \frac{\beta}{2} l_s, \quad (11)$$

where $\alpha$ and $\beta$ are Writefull, and $BCELoss(\cdot)$ is the binary cross-entropy loss [51]. The target domain federated GAT training with the expanded graph following the HFL paradigm.

**Target model fine-tuning** After target domain training with the expanded graph, the target domain GAT model assimilates knowledge from the source domains. However, NT may still be unavoidable, potentially leading to the accumulation of negative knowledge in the target domain. An example of this is the Gaussian noise matrices transferred from source domains where the user has no interactions. On the basis of this consideration, we adopt an additional fine-tuning stage: First, we freeze the message propagation layers of GAT to isolate the influence of source domains preventing negative information from permeating through the transfer process. Second, we directly train the well-informed embeddings generated by the target domain GAT. Adapting the learned external knowledge through these steps enables more accurate prediction of ratings in the target domain. In this process, we use the loss of prediction in Equation (11) as the object function:

$$L_{ft} = BCELoss(\hat{\mathbf{R}}_{uv}^T, \mathbf{R}_{uv}^T). \quad (12)$$

We provide a computational analysis and a communication analysis of FedGCDR in Appendix B.

## 4 Experiments

### 4.1 Experimental setup

**Datasets** We study the effectiveness of FedGCDR with 16 popular domains of a real-world dataset **Amazon** [52]. To study the impact of the number of domains on model performance, we divide these

Table 1: Statistics on the Amazon Dataset. (min-median-max) values are provided for $|U_d|$, $|I_d|$ and $|R_d|$.

| Dataset | $|U|$ | $|U_d|$ | $|I_d|$ | $|R_d|$ | avg (sparsity) |
|---|---|---|---|---|---|
| Amazon-4 | 55,518 | 6,632 - 12,626 - 27,402 | 53,082 - 134,438 - 501,153 | 623,420 - 646,266 - 5,481,801 | 0.0802% |
| Amazon-8 | 99,506 | 6,632 - 13,978 - 27,402 | 53,082 - 106,985 - 501,153 | 186,016 - 618,539 - 5,481,801 | 0.0399% |
| Amazon-16 | 117,672 | 1,036 - 9,038 - 27,402 | 17,209 - 64,624 - 501,153 | 41,427 - 379,657 - 5,481,801 | 0.0928% |
| Amazon-Dual | 2,500 | 2,500 - 2,500 - 2,500 | 17,889 - 28,649 - 39,510 | 106,741 - 128,601 - 150,461 | 0.1955% |

domains into three subsets containing 4, 8, and 16 domains respectively and denote them as **Amazon-4**, **Amazon-8**, and **Amazon-16** respectively. We randomly selected 2500 overlapping users in the Books domain and CDs domain to construct the dataset **Amazon-Dual** to validate the performance of our FedGCDR in the conventional dual-domain scenarios where users full-overlap.The statistics of sub-datasets are shown in Table 1. We filter the original data in different ways, and more details are given in Appendix C.1. In our experiments, Books and CDs are selected as the target domains. For the ratings in each domain, we first convert them to implicit data, where entries corresponding to existing user-item interactions are marked as 1 and others are marked as 0.

**Baselines**   We compare FedGCDR with the following state-of-the-art models: (1) **FedGNN** [34] is an attempt to adopt FL graph learning to recommender systems. Its recommendation performance could represent the data quality of the target domain and reflect negative transfer. In Tables 2 and 3, in order to distinguish **FedGNN** from the CDR baselines, we denote it by **Single Domain**. (2) **EMCDR** [15] is a conventional embedding-mapping CDR framework. We adjust it to the HFL framework following [6]. (3) **PriCDR** [19] is a privacy-preserving CDR framework, which adopted DP on the rating matrices to ensure privacy. (4) **FedCT** [6] is a VAE-based federated framework that is the first attempt to protect intra-domain privacy in cross-domain recommendations. (5) **FedCDR** [7] is a dual-target federated CDR framework, where the user embeddings are transferred as knowledge to enhance the other domain's model training. To adapt to the BS-CDR scenarios, we modify **FedCDR** by applying embedding averaging when receiving source domain embeddings.

We provide implement details in Appendix C.2.

## 4.2   Recommendation performance

We report the model performance results in Tables 2 and 3. **Single domain** shows that the Book domain has better single-domain recommendation accuracy than the Music domain, which represents higher data quality and quantity. Under BS-CDR settings, **FedGCDR** outperforms all CDR baselines on all three sub-datasets, which confirms the effectiveness of the proposed model on real-world data.

To further study our model capacity in alleviating negative transfer, we first define two types of negative transfer: (1) Soft Negative Transfer (SNT), where recommender models' performance under the multi-domain setting is worse than that under the single-domain setting. This means that the knowledge from source domains poisoning the target domain's model training. (2) Hard Negative Transfer (HNT), where recommended performance of a large number of source domains is lower than that of a small number of source domains. This means that the newly added domains are not conducive to the training of the target domain or conflict with the already added source domain.

Taking the Books domain as the target domain, **EMCDR**, **PriCDR**, **FedCT** and **FedCDR** both have serious negative transfer problems and lower performance on the three data subsets. From the SNT perspective, their performances is much worse than that of **Single Domain** as shown in Figure 4. From the HNT perspective, their performances under 16-domain settings is worse than that under the 8-domain and 4-domain settings, which suggests it is not appropriate to recklessly transfer knowledge to a well-informed domain. Our **FedGCDR** model successfully alleviates NT with consistently best and stable performance results. For the CDs domain, the performance of the CDR models greatly improves with less NT in Figure 4. From the SNT perspective, information-poor domains have a lower probability of negative transfer, as they are inherently less well-trained. From the HNT perspective, on the Amazon-8 dataset, the performance of all models declines, which we attribute to the strong negative knowledge introduced by the four additional domains compared to the Amazon-4 dataset. On the Amazon-16 dataset, all methods achieve best performance which indicates

Table 2: The recommendation performance on Amazon@Books. Single Domain represents FedGNN and its performance is exactly the same on three sub-datasets. FedGCDR-DP is a complete implementation of our method while FedGCDR does not incorporate Gaussian noise. (The best result for the same setting is marked in bold and the second best is underlined. These notes are the same to others.)

| Model | Amazon-4@Books | | | | Amazon-8@Books | | | | Amazon-16@Books | | | |
|---|---|---|---|---|---|---|---|---|---|---|---|---|
| | HR@5 | NDCG@5 | HR@10 | NDCG@10 | HR@5 | NDCG@5 | HR@10 | NDCG@10 | HR@5 | NDCG@5 | HR@10 | NDCG@10 |
| Single Domain | 0.4693 | 0.3188 | 0.6067 | 0.3634 | 0.4693 | 0.3188 | 0.6067 | 0.3634 | 0.4693 | 0.3188 | 0.6067 | 0.3634 |
| EMCDR | 0.4633 | 0.3075 | 0.6179 | 0.3191 | 0.4678 | 0.3268 | 0.5990 | 0.3518 | 0.3140 | 0.2184 | 0.4207 | 0.2348 |
| PriCDR | 0.4061 | 0.3159 | 0.5275 | 0.3550 | 0.4409 | 0.3196 | 0.5913 | 0.3681 | 0.3699 | 0.2650 | 0.4914 | 0.3042 |
| FedCT | 0.2911 | 0.2044 | 0.4276 | 0.2482 | 0.4665 | 0.3516 | 0.6002 | 0.3939 | 0.2779 | 0.2335 | 0.3580 | 0.2593 |
| FedCDR | 0.4115 | 0.3153 | 0.5415 | 0.3570 | 0.4791 | 0.3538 | 0.6182 | 0.3967 | 0.3926 | 0.2907 | 0.5626 | 0.3403 |
| FedGCDR-DP | 0.4903 | 0.3417 | 0.6717 | 0.3733 | 0.5224 | 0.3608 | 0.6727 | 0.3973 | 0.4928 | 0.3509 | 0.6510 | 0.3742 |
| FedGCDR | **0.4941** | **0.3592** | **0.6732** | **0.3920** | **0.5300** | **0.3686** | **0.6752** | **0.3985** | **0.5016** | **0.3600** | **0.6516** | **0.3854** |

Table 3: The recommendation performance on Amazon@CDs.

| Model | Amazon-4@CDs | | | | Amazon-8@CDs | | | | Amazon-16@CDs | | | |
|---|---|---|---|---|---|---|---|---|---|---|---|---|
| | HR@5 | NDCG@5 | HR@10 | NDCG@10 | HR@5 | NDCG@5 | HR@10 | NDCG@10 | HR@5 | NDCG@5 | HR@10 | NDCG@10 |
| Single Domain | 0.4119 | 0.2751 | 0.5031 | 0.3040 | 0.4119 | 0.2751 | 0.5031 | 0.3040 | 0.4119 | 0.2751 | 0.5031 | 0.3040 |
| EMCDR | 0.4074 | 0.2651 | 0.5591 | 0.2972 | 0.2882 | 0.1828 | 0.4361 | 0.2199 | 0.4704 | 0.3683 | 0.5740 | 0.3937 |
| PriCDR | 0.3987 | 0.2838 | 0.5114 | 0.3202 | 0.2946 | 0.1988 | 0.4229 | 0.2400 | 0.4405 | 0.3689 | 0.5399 | 0.4011 |
| FedCT | 0.2681 | 0.1603 | 0.3774 | 0.1956 | 0.1801 | 0.1282 | 0.3001 | 0.1681 | 0.3522 | 0.2963 | 0.4326 | 0.3219 |
| FedCDR | 0.4299 | 0.2949 | 0.5636 | 0.3381 | 0.3088 | 0.2109 | 0.4620 | 0.2600 | 0.4823 | 0.3983 | 0.5808 | 0.4297 |
| FedGCDR-DP | 0.4359 | 0.2960 | 0.5779 | 0.3520 | 0.4122 | 0.2983 | 0.5064 | 0.3106 | 0.4963 | 0.4061 | 0.6135 | 0.4453 |
| FedGCDR | **0.4588** | **0.3282** | **0.5819** | **0.3679** | **0.4276** | **0.3142** | **0.5270** | **0.3464** | **0.5267** | **0.4382** | **0.6208** | **0.4684** |

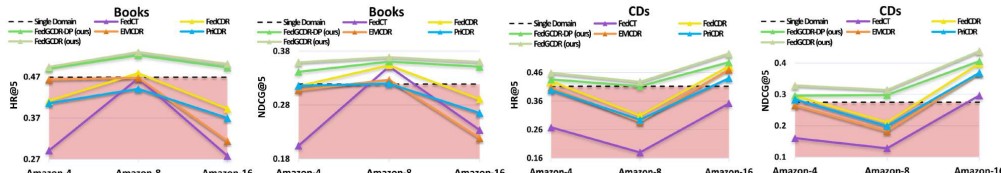

Figure 4: Illustrations of negative transfer on HR@5 and NDCG@5. Metric values lower than single-domain (dotted line and red area) mean severe negative soft negative transfer. The figure on HR@10 and NDCG@10 is shown in Appendix D.1.

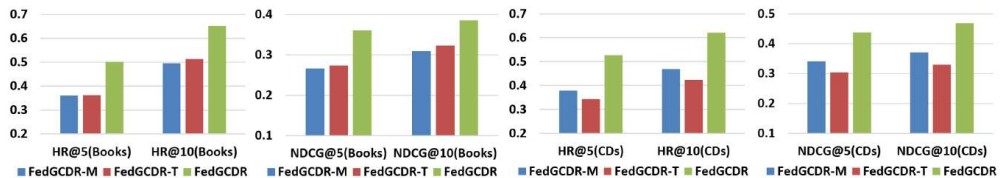

Figure 5: Ablation study on Amazon-16@CDs and Amazon-16@Books.

more knowledge from the source domains can improve the model performance in the CDs domain. Overall, the capability of **EMCDR**, **PriCDR** and **FedCDR** to alleviate negative transfer is much higher than that of **FedCT**. This is because the proportion of target domain features in the final feature is guaranteed by tuning Writefull to control the transfer ratio of the source domain. Meanwhile, **FedGCDR** avoids this kind of human involvement and maintains performance optimality on three sub-datasets. In conclusion, our experiments show the superiority of **FedCDR** in recommendation performance and the effectiveness of alleviating NT.

### 4.3 Ablation study

To study the contribution of each module of **FedGCDR**, we implement two model variants, **FedGCDR-M** and **FedGCDR-T**. **FedGCDR-T** transfers the source domain embeddings with-

Table 4: Dual-domain CDR performance.

| Model | Books → CDs | | Books ← CDs | |
|---|---|---|---|---|
| | HR@10 | NDCG@10 | HR@10 | NDCG@10 |
| Single Domain | 0.2713 | 0.1429 | 0.2594 | 0.1524 |
| EMCDR | 0.2816 | 0.1409 | 0.2596 | 0.1540 |
| PriCDR | 0.2903 | 0.1446 | 0.2662 | 0.1583 |
| FedCT | 0.2384 | 0.1239 | 0.2570 | 0.1551 |
| FedCDR | 0.2566 | 0.1376 | 0.2657 | 0.1554 |
| FedGCDR-DP | 0.3076 | 0.1552 | 0.2749 | 0.1602 |
| FedGCDR | **0.3323** | **0.1838** | **0.2958** | **0.1797** |

out mapping. **FedGCDR-M** replaces the attention graph expansion with the average sum of source domain embeddings and omits the fine-tuning stage. We experiment with Books and CDs as target domains on the Amazon-16 dataset. The experimental results are shown in Figure 5. We make the following observations: (1) The two variants perform differently on different target domains. On the Books domain, **FedGCDR-T** performs better than **FedGCDR-M**, which indicates that for domains with higher data quality, preventing the transfer of negative knowledge from other domains is more important than mapping this knowledge better (in other words, the quality of external information holds greater significance than its quantity), and the Positive Knowledge Activation module meets the requirements of such domains. On the CDs domain, **FedGCDR-M** performs better than **FedGCDR-T**, which indicates that for domains that are deficient in information, mapping knowledge correctly is more important than preventing inter-domain negative knowledge (in other words, the quantity of external information holds greater significance than its quality), and the Positive Knowledge Transfer module meets these requirements. (2) Compared to **FedGCDR**, the absence of either module can cause a significant drop in performance. This indicates that in cross-domain recommendation, we should not only focus on transferring positive knowledge, but also control the spread of negative knowledge to the target domain, especially when a large number of domains.

### 4.4 Dual-domain scenario

According to the experimental results shown in Table 4, our **FedGCDR** achieved the best experimental metrics in both knowledge transfer directions. This shows that our approach is also suitable for dual-domain scenarios where users full-overlap and have only a single source domain and a single target domain.

We provide experimental results on privacy budget in Appendix D.2.

## 5 Limitations

Our experiments were conducted on 16 domains of the Amazon dataset. While this extensive dataset covers broader source domains, relying on a single dataset may limit the generalizability of our model to data from other sources. Our approach uses overlapping users as a cross-domain bridge. Indeed, there are no widely-recognized cross-domain recommendation datasets with more than three domains, aside from the Amazon dataset. Despite this limitation, we believe that the improvements in privacy preservation and model performance demonstrated by FedGCDR underscore its superiority.

## 6 Conclusion

We proposed FedGCDR, a federated graph learning framework designed for BS-CDR. FedGCDR addresses the critical challenge of privacy preservation and negative transfer by employing a positive knowledge transfer module and a positive knowledge activation module. Our method achieves best recommendation quality results on 16 domains of the Amazon dataset. In the future, we aim to extend FedGCDR to improve the recommendation performance of both the target and the source domains.

## 7 Acknowledgment

The research was supported by Natural Science Foundation of China (62272403).

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

# A  Privacy analysis

Due to the algorithmic nature of GNN, the source domain embeddings we pass are a function result on the user embeddings and item embeddings. This means that in the event of a successful inference attack, our user item interaction matrix is exposed to the threat of privacy disclosure. We apply differential privacy (DP) to further safeguard embeddings, following the approach of Cai et al. [53].

**Threat model**   In this paper, we assume the threat model to be semi-honest (honest-but-curious). Under this threat model, the participants adhere strictly to the FL protocol for collaborative model training. However, they are interested in the sensitive rating data and may attempt to extract as much information as possible from the transferred embeddings. Specifically, these semi-honest parties, i.e. the target domain, may employ inference attacks [54] on the embeddings to reconstruct or infer sensitive user-item interaction matrix of other domains.

DEFINITION 1 (THE GAUSSIAN MECHANISM). *Given a function $f : D \to \mathbb{R}^d$ over a dataset D, the Gaussian mechanism is defined as*:

$$F_G(x, f(\cdot), \epsilon) = f(x) + (r_1, ...r_k),$$ (13)

where $r_i$ is the random noise drawn from $\mathcal{N} \sim (0, \sigma^2 \Delta_2 f^2)$ and $\sigma = \frac{\sqrt{2ln(1.25/\delta)}}{\epsilon}$. In FedGCDR, the intra-domain GAT-based federated model is considered as the function $f(\cdot)$.

THEOREM 2. *The Gaussian mechanism defined in Definition 1 preserves $(\epsilon, \delta)$-DP for each publication step [22].*

First, we give the definition of the inverse function:

DEFINITION 2 (INVERSE FUNCTION). *Given a function $f : D \to \mathbb{R}^d$ over a dataset D, the inverse function $f^{-1}$ is defined as*:

$$f^{-1} = argmin_g \sum_{i \in u \bigcup v} \| \mathbf{e}_i - g(f(\mathbf{e}_u, \mathbf{e}_v)) \|_2,$$ (14)

where

$$\mathbf{e}_u, \mathbf{e}_v = Embedding(x), x \in D.$$ (15)

For the target domain, the embeddings received form source domains can be regarded as the functional result of their models. Let the function be $f(\cdot, \cdot)$ and the input $e_u, e_v$ is the user embedding and item embedding respectively. The embeddings is the output $f(e_u, e_v)$. The target domain attempts to find a inference attack function $I(\cdot)$ which is as close to the inverse function as possible.

DEFINITION 3 (PRIVACY LEAKAGE). *Given a function $f : E \to \mathbb{R}^d$ over a Embedding set E and an inference function I, the privacy leakage $\Lambda$ is defined as*:

$$\Lambda = \frac{1}{1 + \frac{1}{|U|} \sum Leak_u + \frac{1}{|V|} \sum Leak_v}.$$ (16)

where

$$Leak_u = \| \mathbf{e}_i - I_u(f(\mathbf{e}_u, \mathbf{e}_v)) \|_2, i \in U,$$ (17)

$$Leak_v = \| \mathbf{e}_j - I_v(f(\mathbf{e}_u, \mathbf{e}_v)) \|_2, j \in V.$$ (18)

$\| e - I(f(e_u, e_v)) \|_2$ reflects the closeness of the reconstructed input to the true input. Therefore, privacy leakage (PLeak) $\Lambda$ is able to reflect privacy leakage of FedGCDR with the inference function $I(\cdot)$. PLeak equal to 1 means a perfect reconstruction, and being close to zero means a bad reconstruction. DP on the embeddings further ensures that attackers cannot perfectly reconstruct the raw data.

THEOREM 2. *If PLeak equals to 1 with the inference function $I(\cdot)$, the function $f(\cdot, \cdot)$ is bijection.*

*Proof.* If the function $f(\cdot, \cdot)$ is not bijection, there are $i, j \in D$ and $i \neq j$, but $f(\mathbf{e}_u^i, \mathbf{e}_v^i) = f(\mathbf{e}_u^j, \mathbf{e}_v^j)$ and $I(f(\mathbf{e}_u^i, \mathbf{e}_v^i))$ and $I(f(\mathbf{e}_u^j, \mathbf{e}_v^j))$. This is a contradiction as the perfect reconstruction requires both $\mathbf{e}_u^i, \mathbf{e}_v^i = I(f(\mathbf{e}_u^i, \mathbf{e}_v^i))$ and $\mathbf{e}_u^j, \mathbf{e}_v^j = I(f(\mathbf{e}_u^j, \mathbf{e}_v^j))$ to achieve $\Lambda = 1$. Therefore, the function $f(\cdot, \cdot)$ must be bijection.

THEOREM 3. *Given the lipschiz constant L of the function f at $x \in D$ with the noise generated by Gaussian mechanism $\mathcal{N}$ on embeddings. If $f(\mathbf{e}_u, \mathbf{e}_v) + \mathcal{N} \in f$, the distance between $x$ to the reconstructed data of the attack $I(\cdot)$ which achieves $\Lambda = 1$ is bounded by $\frac{|\mathcal{N}|}{L}$.*

*Proof.* By Theorem 2, we have for $x \in D$ and $v \in f, I(f(\mathbf{e}_u, \mathbf{e}_v)) = (\mathbf{e}_u, \mathbf{e}_v)$ and $f(I(v)) = v$, From the Lipschitz continuous,

$$|e - I(f(\mathbf{e}_u, \mathbf{e}_v) + \mathcal{N})| \geq \frac{|f(\mathbf{e}_u, \mathbf{e}_v) - (f(\mathbf{e}_u, \mathbf{e}_v) + \mathcal{N})|}{L} = \frac{|\mathcal{N}|}{L}. \tag{19}$$

Therefore, by perturbing the source domain embedding with Gaussian mechanism, the reconstructed data of the ideal attack deviates from the real data and prevents a perfect reconstruction (*i.e.*, $\Lambda = 1$).

## B  Cost analysis

Due to the complexity of the FedGCDR pipeline, we perform a theoretical analysis of the computational and communication cost of FedGCDR accordance with the HVH pipeline including horizontal source domain training, vertical positive knowledge transfer module and horizontal positive knowledge activation module.

### B.1  Computational cost

Given a GAT model, let $V$ be the number of nodes, $E$ be the number of edges, and $F$ the embedding size. The computational cost of one propagation layer of classic GAT framework is $O(VFF' + EF')$ [38]. In the horizontal source domain training, our model is a simplified GAT variants which discard feature transformation and non-linear activation. For a $N^K$ layer model, the simplified computational cost is $O(N^K EF)$. In the vertical positive knowledge transfer module, space mapping is carried by a $N^m$ layers' MLP with computational cost $O(N^m F^2)$. In the horizontal positive knowledge activation module, the first part is the simplified GAT model and the second part is the fine-tuning model with computational cost $O(F^2)$. In conclusion, for the FedCDR framework with $N^{\mathcal{D}}$ domains, $T^G$ GAT-based federated model training epochs, and $T^F$ fine-tuning epochs, the total computational cost is $O(T^G(N^{\mathcal{D}} N^K EF + N^m F^2) + T^F F^2))$. Cause $N^m F^2 \ll N^{\mathcal{D}} N^K EF$, we get the final computational cost $O(T^G N^{\mathcal{D}} N^K EF + T^F F^2))$.

### B.2  Communication cost

In FedGCDR, the global model and item embeddings are held by the domain server. Let $I$ be the number of items and $F$ be the embedding size. The space complexity of global model and item embeddings are $O(F)$ and $O(IF)$ respectively. In the horizontal source domain training, the domain server distributes the global model and item embeddings and get the gradient with the same size. The communication cost is $O(F + IF)$. In the vertical positive knowledge transfer module, the $N^m$ layers' MLP and its gradients are transmitted with the communication cost $O(N^m F^2)$. In the horizontal positive knowledge activation module, the target domain additionally perform a fine-tuning stage with communication cost $O(IF)$. In conclusion, for the FedCDR framework with $N^u$ users, $T^G$ federated model training epochs, and $T^F$ fine-tuning epochs, the total communication cost is $O(T^G N^u(N^m F^2 + F + IF) + T^F N^u IF)$. Cause $N^m F^2 + F \ll IF$, we get $O(N^u IF(T^G + T^F))$. According to the expression, the communication cost of our FedCDR is basically equivalent to the cost of two HFL progress. The cost is reduced because knowledge transfer totally takes place in user space, thus avoiding large-scale information exchange.

## C  Experimental details

### C.1  Dataset details

The Amazon dataset we used is the 2018 version and can be easily accessed in `https://cseweb.ucsd.edu/~jmcauley/datasets/amazon_v2/`. The basis of our domain selection strategy is the amount of data before performing data filtering. Thus, we sorted the domains contained in the Amazon dataset based on the amount of data in descending order and selected the top 16 domains. Similarly, the Amazon-4 and Amazon-8 datasets were selected accordingly. The only exception is

that we prioritized the Movie domain, which has a relatively small amount of source data, based on popularity. In addition to multi-domain experiments, we randomly selected 2500 overlapping users in the Books domain and CDs domain to construct the dataset **Amazon-Dual**, so as to validate the performance of our FedGCDR in the conventional dual-domain scenarios where users full-overlap. The processing details are shown in Table 5. The bottleneck time of FedGCDR is the federated-GAT training time in each domain, and we also show it in Table 5.

Table 5: Processing details on Amazon Dataset.

| Domain | $|U|$ | $|I|$ | user core | item core | time per epoch (mm:ss) | |
|---|---|---|---|---|---|---|
| Clothing, Shoes and Jewelry | 11,558 | 197,677 | 24 | 10 | 03:31 | |
| Books | 27,402 | 501,153 | 96 | 10 | 11:02 | Amazon-4 |
| CDs and Vinyl | 13,694 | 71,199 | 24 | 10 | 04:02 | |
| Movies and TV | 6,632 | 53,082 | 48 | 10 | 01:55 | |
| Home and Kitchen | 15,772 | 135,182 | 48 | 10 | 04:36 | |
| Electronics | 16,836 | 120,876 | 32 | 10 | 04:40 | Amazon-8 |
| Sports and Outdoors | 14,262 | 93,095 | 32 | 10 | 03:41 | |
| Cell Phones and Accessories | 9,312 | 55,312 | 24 | 10 | 02:40 | |
| Tools and Home Improvement | 9,899 | 65,378 | 16 | 10 | 02:47 | |
| Toys and Games | 5,267 | 63,870 | 32 | 10 | 01:10 | |
| Automotive | 6,135 | 62,188 | 32 | 10 | 01:32 | |
| Pet Supplies | 4,280 | 31,853 | 32 | 10 | 00:55 | |
| Kindle Store | 8,756 | 82,874 | 48 | 10 | 02:06 | Amazon-16 |
| Office Products | 1,266 | 17,209 | 32 | 10 | 00:16 | |
| Patio, Lawn and Garden | 1,036 | 17,605 | 32 | 10 | 00:16 | |
| Grocery and Gourmet Food | 3,415 | 36,292 | 32 | 10 | 00:50 | |

## C.2 Implement details

We provide the implemented details of our proposed model and baselines. We set batch size = 256 and latent dim = 8 for all domains. The number of propagation layer of GAT-base federated model is set to 2. The MLP has two hidden layers with size={16, 4}.Considering the trade-off between recommendation performance and privacy preservation, we set $\epsilon$ to 8 and $\sigma$ to $10^{-5}$. We set $\alpha$=0.01 and $\beta$=0.01 which are the two Writefull of the objective function $L_{GAT}(\cdot)$. When training our models, we choose Adam as the optimizer, and set the learning rate to 0.01 both in GAT-based federated model training and the fine-tuning stage. To evaluate the recommendation performance, we use the leave-one-out method which is widely used in recommender systems [51]. Specifically, we held out the latest interaction as the test set and utilized the remaining data for training. Then, we follow the common strategy which randomly samples 99 negative items that are not interacted with by the user for the rank list generation of the test set. We consider the top-$k$ recommendation task as the main experiment so we choose metrics including Hit Ratio (HR)@K score and the Normalized Discounted Cumulative Gain (NDCG)@K [55] of the top-K ranked items with K=5, 10. We conduct the experiments on three groups of random seeds and report the average results. We conduct all the experiments on NVIDIA 3090 GPUs.

# D Additional experimental results

## D.1 Neagtive transfer on HR@10 and NDCG@10.

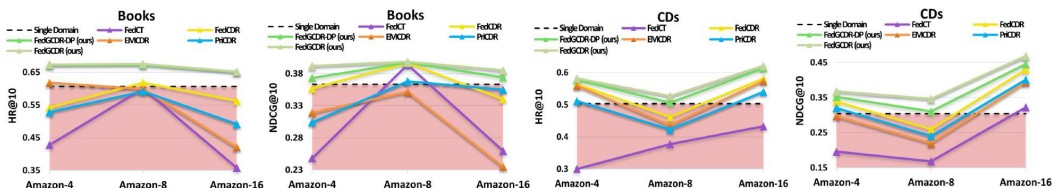

Figure 6: Illustrations of negative transfer on HR@10 and NDCG@10

For HR@10 and NDCG@10 in Figure 6, our method and baselines show similar trends to the previous HR@5 and NDCG@5. Compared to Figure 6, the slight difference is that FedCT's HR@10

performance is better on Amazon-8@CDs than on Amazon-4@CDs. We believe that the reason is the poor performance of FedCT on Amzon-4@CDs lowers the threshold for negative transfer of the newly added source domain.

## D.2 Privacy budget

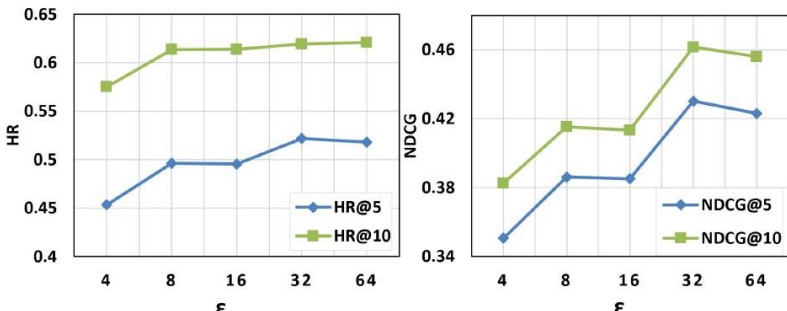

Figure 7: The effect of $\epsilon$ in DP on model performance.

To study the effects of privacy budget $\epsilon$ on the model performance, we vary the privacy budget $\epsilon = \{4, 8, 16, 32, 64\}$ to affect the $\sigma$. We experimented on the Amazon-16 with CDs as the target domain and fix $\delta = 10-5$. We report the results Figure 7. From that we can observe that the model's performance decreases as $\epsilon$ decreases. The degradation in model performance suggests that our approach struggles to counteract the effects of high-intensity noise in a large number of domains, but the model performance is not completely destroyed by Gaussian noise. Thus, there is a trade-off between accuracy and privacy, where a smaller $\epsilon$ value adds more noise to embeddings for stronger privacy preservation but leads to more prediction error. Therefore, to balance the data privacy preservation capacity and the model performance, we set it as $\epsilon = 8$.

# E Broader impacts

Our proposed FedGCDR is tailored for BS-CDR, focusing on the privacy and negative transfer problems. CDR is widely uesd, while BS-CDR is generic and close to the reality. Our approach can better mine user preferences and effectively protect privacy. On the one hand, users will benefit from more accurate recommendations and thus have a better experience in shopping, watching movies, etc. On the other hand, various economic entities can gain more profits.

