# OpenReview forum: "Federated Graph Learning for Cross-Domain Recommendation"
_NeurIPS.cc/2024/Conference — NeurIPS 2024 poster_

### Official Review · Reviewer_yU7k · 2024-07-08

**Soundness:** 4
**Presentation:** 4
**Contribution:** 4
**Rating:** 7
**Confidence:** 5

**Summary:**

This paper introduces an innovative federated CDR framework with two key modules tailored for privacy preserving and negative transfer. For privacy, it presents a solid theoretical guarantee. For negative transfer, it generates domain attentions by virtual social links and conduct a fine-tuning stage to filter source domain knowledge. With the comprehensive empirical evaluation, it shows the superiority of the proposed framework.

**Strengths:**

1. Innovation: Utilizing HVH structure and Gaussian noise to ensure all-round privacy, GAT with virtual links to generate domain attentions and comprehensive objective function with prediction loss, mapping loss and social regularization loss seems technically sound.
2. Theoretical analysis: The paper provides a robust theoretical analysis. The detailed explanation concerning protecting transferred high-order embeddings from inference attacks strengthen significantly strengthen the paper's theoretical framework.
3. Generally, the proposed method is complex but this paper is well-organized and easy to follow.
4. Experimental demonstrations: The authors have conducted extensive experiments to validate the effectiveness of the proposed method, including the performance comparison, the ablation study, the privacy budget study and an additional dual-domain study. The compare of overall performance validates the superiority of proposed model. The ablation study validates the effectiveness of two key modules and the privacy budget study demonstrate the balance of privacy preservation and model performance. The dual-domain study proves that the proposed model can also cope with traditional scenarios.

**Weaknesses:**

1. In Figure 2, the use of the terms "extended" and "expansion" appears to be inconsistent.  Consistent use of terminology throughout the paper will aid in the reader's comprehension and the overall clarity of the presentation
2. The authors have conducted extensive experiments on the Amazon dataset as detailed in the experimental section. However, as the authors themselves acknowledge in the limitation section, the generalization performance of their framework has not been sufficiently validated. Given this, and despite the fact that the experiments may not align with the authors' definition of BD-CDR, it is recommended that the authors consider conducting experiments on other datasets (e.g., Douban) to further verify the generalizability of their framework. This additional testing would strengthen the paper's claims regarding the robustness of the proposed approach.

**Questions:**

1. From the perspective of privacy, DP is widely used in previous method. What are the privacy innovations in this paper? What is the unique technical contribution between this work and existing works on privacy-preserving CDR?

**Limitations:**

Please refer to weaknesses.

---

> ### Author Rebuttal · Authors · 2024-08-06
>
> We would like to sincerely thank you for positive evaluations and valuable comments for improvement.
>
> **W1:** The use of the terms "extended" and "expansion" appears to be inconsistent.
>
> **Response:** We agree and **we will use *expand/expansion* uniformly in paper’s updated version.**
>
> **W2:** The generalization performance of the framework has not been sufficiently validated.
>
> **Response:** We agree that it is important to verify the generalization performance of the model on other datasets. Thus, we conduct new experiments on Douban dataset to validate the generalization capability of the proposed method.  As shown in Table u2 in **Global Rebuttal** pdf file, our method yields both optimal and sub-optimal results on almost all metrics. **We will add these new results in the updated version**. For details, please refer to the **Global Rebuttal** with pdf attachment.
>
> **Q1:**  What are the privacy innovations in this paper?
>
> Response: We provide a full range of privacy both intra-domain and inter-domain. In our work, we include two horizontal federation phases and DP (line 177, **Privacy-preserving knowledge extraction**)  to fully guarantee intra-domain privacy and inter-domain privacy. In previous work, those focusing on firm-level privacy tend to protect inter-domain privacy through privacy techniques (DP, projection methods, etc.); those focusing on user-level privacy tend to consider only intra-domain privacy (embeddings from different domains are passed in plaintext). In the **Appendix** of paper's previous version, we had provided a theoretical proof of privacy preservation (**Appendix A**) and experimentally verify the model performance under different privacy budgets (**Appendix D.3**). Our approach considers the full range of privacy protection, which distinguishes us from previous approaches, and outperforms all baselines.

---

### Official Review · Reviewer_xoiK · 2024-07-11

**Soundness:** 3
**Presentation:** 3
**Contribution:** 3
**Rating:** 8
**Confidence:** 5

**Summary:**

This paper presents a novel federated framework of CDR, FedGCDR, for privacy preserving and negative transfer between domains. Its key strengths include a solid theoretical foundation analyzing the DP-based privacy preserving and a novel and dynamic attention generation method to mitigate negative transfer. By tackling both privacy concern and potential negative transfer problem, this work makes a valuable contribution to the field.

**Strengths:**

1. Tackling two Critical Issues for cross-domain recommendation: The paper addresses crucial challenges for privacy preservation and potential negative transfer. By following a horizontal-vertical-horizontal FL pipeline and adopting the Gaussian mechanism, the paper ensures the privacy of both intra-domain and inter-domain. By the graph expansion, the paper generates domain attention to filter harmful info from multiple source domains. Addressing these two issues is essential for fostering a wide range of participation and maintaining a healthy balance between domain involvement and model performance, ultimately contributing to the overall sustainability and scalability of CDR.

2. Novel Approach: The proposed FedGCDR approach is novel in several aspects. First, the problem formulation itself, which focuses on mitigating negative transfer under the privacy constraint, is a departure from existing methods that directly transfer with the consumption of data sparsity and well-chosen domains. Second, the theoretical analysis of the reliability of the Gaussian mechanism protecting the high-order embeddings provides valuable insights and paves the way for a secure transfer between domains. Third, the incorporation of GAT as a tool for both mining domain knowledge and generating dynamic domain attention to filter potentially harmful information is a novel method to address the negative transfer issue, which has been a long-standing challenge in CDR.

3. Theoretical Guarantees and Practical Considerations: The paper presents a solid theoretical foundation by analyzing the Gaussian mechanism to protect high-order embeddings output by GAT. Furthermore, by combining horizontal fl with vertical fl, knowledge transfer down to edge devices is realistic and avoids the additional overhead required for the vertical process.

4. Comprehensive Empirical Evaluation: The paper's strength lies in its comprehensive empirical evaluation across sixteen widely used domains of the Amazon dataset, spanning various sub-datasets of different domain numbers. Besides the overall model performance comparison, the paper compares the ability of mitigating negative transfer with defined concept ‘soft negative transfer’ and ‘hard negative transfer’. The target domain setting with different data quality further demonstrates the generalization of the model and the irrationality of the direct transmission of the previous method.

**Weaknesses:**

1. The paper addresses the broader-domain CDR involving more than three domains. While multi-domain CDR is undoubtedly significant and has been the subject of extensive research, the emphasis on the number "three" is noted. Could the authors clarify the significance of this number in their work? Does this particular emphasis distinguish their research from existing studies? If not central to the research, it is suggested that the authors reconsider the emphasis on this number to avoid potential misunderstandings or unnecessary focus.

2. By introducing multiple social regulation terms, yet the authors have chosen one specific term (as shown in Equation 9). It would be beneficial for the authors to provide an explanation for this selection. Is this choice based on theoretical analysis, empirical results, or comparative study with existing literature? Clearly articulating the basis for this choice will help readers understand its importance and applicability.

**Questions:**

1. This article is mainly from the user's perspective to solve the privacy problem and whether the proposed framework can be extended to more realistic company-level applications.

2. Negative transfer is an important problem in CDR as well as transfer learning. However, facing the ‘when to transfer’ problem [1] in CDR, it seems to be a potential method to stop the negative transfer when it occurs. Why cannot we adopt this simple method to avoid such complex attention generation progress?

[1] Pan S J, Yang Q. A survey on transfer learning[J]. IEEE Transactions on knowledge and data engineering, 2009, 22(10): 1345-1359.

**Limitations:**

Please refer to the weak points.

---

> ### Author Rebuttal · Authors · 2024-08-06
>
> We sincerely thank you for your valuable comments and suggestions. We hope our response addresses your concerns.
>
> **W1:** Why addresses the broader-domain CDR involving more than three domains.
>
> **Response:** Because solving the cross-domain recommendation problem with more than three participating domains is very important for better mining user preferences. In existing work in the area of cross-domain recommendation, researchers tend to focus on only two domains. These works often do not generalize well to multi-domain tasks. There are also some studies that begin to focus on multi-domain knowledge transfer, but they are also limited to only three domains [1]. We believe that these studies do not fully reflect the nature of multi-domain, while paying insufficient attention to privacy and negative transfer challenges. These two challenges become more severe with the number of domains, in line…. We therefore emphasize a number of domains greater than three in the article to distinguish our method from previous works. As shown in Table 1 and Table 2 of paper’s previous version, our methods (FedGCDR, FedGCDR-DP) outperform all baselines under the settings of 4 domains, 8 domains and 16 domains.
>
> [1] Liu W, Chen C, Liao X, et al. Federated Probabilistic Preference Distribution Modelling with Compactness Co-Clustering for Privacy-Preserving Multi-Domain Recommendation[C]//IJCAI. 2023: 2206-2214.
>
> **W2:** Why choose the special social term in Soreg.
>
> **Response:** We use the social term in Equation 9 because it better exploits the user's interests. In Soreg, the authors propose two innovative social terms, as follows:
>
> $(1)\sum_{i=1}^m || U_i -\frac{\sum_{f \in \mathcal{F}^+(i) } Sim(i,f) \times U_f}{\sum_{f \in \mathcal{F}^+(i) } Sim(i,f)}||^2_F$
>
> $(2)\sum^m_{i=1} \sum_{f \in \mathcal{F}^+(i)}Sim(i,f)||U_i-U_f||^2_F$
>
> where, according to Soreg, the second formula: *“is insensitive to users with different tastes”* [2]. For the same user's behavior on different domains, we consider that it shows different dimensions of user interests, which is why we combine as many domains as possible to fully explore user interests. So the same user may exhibit very different tastes on different domains, so we take the more appropriate first social term.
>
> [2] Ma H, Zhou D, Liu C, et al. Recommender systems with social regularization[C]//Proceedings of the fourth ACM international conference on Web search and data mining. 2011: 287-296.
>
> **Q1:**  Whether the proposed framework can be extended to more realistic company-level applications.
>
> **Response:** Yes, our approach fits real-world scenarios and can be modified to better fit company-level scenarios. In real applications, there are many similar setups, e.g., between different recommendation scenarios on online platforms, where their users are partially overlapping and the items are often different. Our assumptions are consistent with the real scenarios, while some modifications to our approach are needed to accommodate company-level applications. In specific, it is necessary to change the intra-domain GAT training and fine-tuning phase of the horizontal FL setting to a centralized training process undertaken by a domain server (e.g., a company). Knowledge transfer between domains (company to company) is still well protected with DP.
>
> **Q2:** Why cannot we adopt stop-transfer method to avoid such complex attention generation progress?
>
> **Response:** Because it's hard to determine whether to stop the transfer when there are multiple source domains. For simpler cross-domain or transfer learning setting, the approach you propose is also really a way to avoid negative transfer.
>
> However, during CDR training, especially in our hypothetical BD-CDR scenario (more than three domains with multiple source domains and one target domain) setup, it would be difficult to determine which source domain or domains lead to negative transfer. Also we need to consider that the time a domain is joined affects whether it produces negative transfer or not. For example, assuming that the knowledge transfer from domain A to domain B is positive, while the re-transfer of knowledge from domain A to domain B, which aggregates the knowledge of domains C and D, is negative. Positive transfer indicates that there is some information or pattern in domain A that is beneficial to domain B. The later negative transfer suggests that after combining the knowledge of domains C and D, the negative knowledge in domain A dominates the transfer process. This shows that it is very arbitrary to rely only on the positive and negative transfer to judge whether to transfer or not, and the correct approach should be for the target domain to actively filter the negative information, so as to fully utilize the knowledge of each domain. Our method filters out potential harmful or conflicting knowledge from source domains, and mitigates the issue of negative transfer.

---

> > ### Comment · Reviewer_xoiK · 2024-08-10
> >
> > Thanks for the detailed explanation. I will keep my score.

---

### Official Review · Reviewer_GMwN · 2024-07-11

**Soundness:** 3
**Presentation:** 3
**Contribution:** 3
**Rating:** 6
**Confidence:** 3

**Summary:**

The paper proposes a novel federated graph learning framework, FedGCDR, aimed at addressing the challenges of privacy and negative transfer (NT) in Broader-Source Cross-Domain Recommendation (BS-CDR) scenarios. The framework includes two key modules: the positive knowledge transfer module and the positive knowledge activation module. These modules ensure privacy preservation and mitigate NT by employing differential privacy and feature mapping techniques, followed by graph expansion and fine-tuning in the target domain. The framework is validated through extensive experiments on the Amazon dataset, demonstrating its superior performance over existing methods.

**Strengths:**

1. The paper considers privacy preservation and negative transfer (NT) challenges under a more generic scenario of Broader-Source Cross-Domain Recommendation (BS-CDR).
2. The framework's design is modular, allowing for easy adaptation and potential integration with other recommendation system components.
3. The experiments cover 16 popular domains from the Amazon dataset and demonstrate that FedGCDR outperforms state-of-the-art methods in terms of recommendation accuracy.

**Weaknesses:**

1. Despite the overall clarity of the writing, there are several clerical mistakes that could easily mislead the reader.
2. While the results on the Amazon dataset are promising, it is unclear how well the model generalizes to other datasets or real-world scenarios with different characteristics.
3. The authors emphasize that the contribution lies in provides high-quality BS-CDR recommendations while safeguarding both user privacy and domain confidentiality. However, the experimental section is completely devoid of any discussion on privacy preservation.

**Questions:**

1. About the clerical error in writing, such as in line 105, should "privacy of individual users (inter-domain privacy)" be "intra-domain privacy"? The line 166 “we learn learning…”, and the line 584 “FedCT’s HR@10 performance is better on Amazon-4@CDs than on Amazon-8@CDs”. But according to Figure 6, it should be the Amazon-8@CDs that outperform the Amazon-4@CDs for FedCT's HR@10.
2. In Section 3.1, the authors assumed that users are partially overlapping between domains, so how do non-overlapping users perform graph expansion in the target domain?
3. The improvement in recommendation performance of the proposed framework is demonstrated in the experimental section, but why another important contribution on privacy preservation is not discussed?
4. Does FedGCDR-DP in Tables 1 and 2 refer to another variant method? What is the difference with the proposed FedGCDR? This is not explained relevantly in the paper.
5. Are the Amazon-4, Amazon-8 shown in Table 4 randomly divided? Or is there another domain selection strategy?

**Limitations:**

Yes. The authors have addressed the limitations.

---

> ### Author Rebuttal · Authors · 2024-08-06
>
> We sincerely thank you for your valuable comments and suggestions. We hope our response addresses your concerns.
>
> **W1&Q1:** Writing errors.
>
> **Response:** Thank you. We agree and **will correct any errors you have raised and scrutinize the paper and fix any typos during the revision process.**
>
> | Line | Writing error | Revised version |
> | --- | --- | --- |
> | 105 | privacy of either individual users (inter-domain privacy) | privacy of either individual users (intra-domain privacy) |
> | 116 | we learn learning a function to estimate the scores | we learn a function to estimate the scores |
> | 584-587 | The slight difference is that FedCT’s HR@10 performance is better on Amazon-4@CDs than on Amazon-8@CDs. | Compared to Figure 6,the slight difference is that FedCT’s HR@10 performance is better on Amazon-8@CDs than on Amazon-4@CDs. |
>
> **W2**: How well does the model generalize to other datasets or real-world scenarios with different characteristics?
>
> **Response**: We agree that it is important to verify the generalization performance of the model on other datasets. Thus, we conduct new experiments on Douban dataset to validate the generalization capability of the proposed method.  As shown in Table u2 in **Global Rebuttal** pdf file, our method yields both optimal and sub-optimal results on almost all metrics. **We will add these new results in the updated version**. For details, please refer to the **Global Rebuttal** with pdf attachment.
>
> **W3&Q3:** The experimental section is completely devoid of any discussion on privacy preservation.
>
> **Response:** In paper’s previous version, we did not detail privacy protection related content in the **experimental section** due to space constraints, instead we added this part in the **Appendix of paper’s previous version**. In accordance with previous work on federated recommendation, we have made full analysis and discussion on privacy protection in both theoretical and experimental aspects. For details, please refer to **Appendix A  Privacy analysis** for theoretical proof and **Appendix D.3 Privacy budget** for the experiments on privacy budget in paper’s previous version.
>
> **Q2:**   How do non-overlapping users perform graph expansion in the target domain?
>
> **Response:** Non-overlapping users perform graph expansion by a set of Gaussian noise matrices. In lines 197 through 198 of paper’s previous version, we had explained the case where there are no corresponding users in one of the source domains in the graph expansion strategy: “*It is worth noting that for source domains where a user has no rating, $X_{S_i}$*  *is a Gaussian noise matrix.”*  Therefore, for non-overlapping users in the target domain, we conduct the graph expansion with $M$ Gaussian noise matrices (assuming $M$ source domains).
>
> **Q4:** Does FedGCDR-DP in Tables 1 and 2 refer to another variant method? What is the difference with the proposed FedGCDR?
>
> **Response:** Yes, FedGCDR and FedCDR-DP are two variants of our proposed method, where FedGCDR does not add Gaussian noise to the knowledge transfer module, while FedGCDR-DP is a complete implementation of the method integrating Gaussian noise. By introducing these two variants it is possible to compare the loss of accuracy due to the addition of inter-domain privacy protection. Thanks for this valuable comment, and **we will add these descriptions to the experimental section in paper’s updated version.**
>
> **Q5:** Domain selection strategy.
>
> **Response:** The basis of our domain selection strategy is the amount of data before performing data filtering. Thus, we sorted the domains contained in the Amazon dataset based on the amount of data in descending order and selected the top 16 domains. Similarly, Amazon-4 and Amazon-8 were selected accordingly. The only exception is that we prioritized the Movie domain, which has a relatively small amount of source data, based on popularity. **We will add this content to the appendix in paper’s updated version.**

---

> > ### Author Response · Authors · 2024-08-13
> >
> > Dear Reviewers,
> >
> > As the deadline for the discussion approaches, we are happy to address any further questions or provide additional clarificaion if needed. Thank you.

---

> > ### Comment · Reviewer_GMwN · 2024-08-13
> >
> > Thank you for your thorough and detailed responses to the reviews. Your explanations have effectively addressed my concerns, particularly regarding the generalization of the model, the domain selection strategy, and the distinctions between the variant methods. I appreciate the additional experiments and clarifications you've included in the updated version of the paper, which have significantly strengthened the overall quality of this work. In light of these improvements, I will be revising my score accordingly.

---

> > > ### Author Response · Authors · 2024-08-14
> > >
> > > Thank you. We appreciate your response, and we are glad that our rebuttal is reassuring.

---

### Official Review · Reviewer_Dn4X · 2024-07-13

**Soundness:** 3
**Presentation:** 3
**Contribution:** 3
**Rating:** 7
**Confidence:** 5

**Summary:**

To solve the privacy issue and negative transfer phenomenon in the cross-domain recommendation, the authors propose a novel framework named FedGCDR. Following the HVH pipeline, two key modules collaboratively transfer positive knowledge and filter the negative interference from source domains. In the experimental part, real world data in sixteen domains validated the model's performance.

**Strengths:**

1.	Novelty. Dynamic attention mechanism in the end-to-end framework ensures an automated domain filtering mechanism. The HVH pipeline makes the training of multi-domain scenes efficient, coupled with differential privacy technology, ensuring the privacy of the training process.
2.	Theory. This paper adopts the Gauss mechanism guarantee on the middle embedding to defend against inference attacks and proves its validity by theory.
3.	Experiments. Based on the Amazon data set, the author carried out a detailed experiment and analyzed the coping ability of various methods to the phenomenon of negative transfer from two perspectives of soft negative transfer and hard negative transfer, thus verifying the superiority of the proposed method. Based on verifying the effectiveness of the two modules, the detailed ablation experiment fully reveals the influence of the two modules on domains of different data quality, which is very convincing.

**Weaknesses:**

1.	The introduction and related works sections of the paper introduce two pivotal concepts: intra-domain privacy and inter-domain privacy. However, the definitions provided for these concepts appear to be inconsistent, which may obscure the paper's motivation regarding privacy concerns. For the paper to effectively convey its contributions and significance, it is crucial that these two concepts are clearly and uniformly defined.
2.	The authors have chosen to emphasize the "Broader-Domain CDR" scenario, which involves more than three domains when defining the problem.  While this distinction is noted, it is not immediately clear why the term "multi-domain CDR" is not used to introduce the framework, as it seems to be a more commonly accepted term in the literature.

**Questions:**

In practical application, heterogeneity exists between different domains, and it may not be appropriate to use the same embedding dimension. Is there a possible solution?

**Limitations:**

Please refer to the weak points.

---

> ### Author Rebuttal · Authors · 2024-08-06
>
> We thank the reviewer for the constructive review and valuable questions that have helped us improve our work.
>
> **W1:** The definitions provided for the intra-domain privacy and inter-domain privacy appear to be inconsistent.
>
> **Response:** We agree and we **will give a unified and complete definition in the introduction section of paper’s updated version.**
>
> **W2:** Why emphasize the concept of  ‘Broader-Domain CDR’.
>
> **Responese:** Because solving the cross-domain recommendation problem with more than three participating domains is very important for better mining user preferences. In existing work in the area of cross-domain recommendation, researchers tend to focus on only two domains. These works often do not generalize well to multi-domain tasks. There are also some studies that begin to focus on multi-domain knowledge transfer, but they are also limited to only three domains [1]. We believe that these studies do not fully reflect the nature of multi-domain, while paying insufficient attention to privacy and negative transfer challenges. These two challenges become more severe with the number of domains (lines 44 through 55). We therefore emphasize a number of domains greater than three in the article to distinguish our method from previous works. As shown in Table 2 of paper’s previous version, our methods (FedGCDR, FedGCDR-DP) outperform all baselines under the settings of 4 domains, 8 domains and 16 domains.
>
> [1] Liu W, Chen C, Liao X, et al. Federated Probabilistic Preference Distribution Modelling with Compactness Co-Clustering for Privacy-Preserving Multi-Domain Recommendation[C]//IJCAI. 2023: 2206-2214.
>
> **Q1:**  Heterogeneity of embedding dimension exists between different domains.
>
> **Response:** For solving the problem of heterogeneity of embedding dimensions, we have two possible approaches. 1. Aligning the attention layers of different domains. Since our method does not directly transmit the original embedding, each domain can choose the appropriate embedding dimension according to its own characteristics. Intermediate embeddings of the same dimension can be obtained by aligning the weight matrices of the attention layers, thus realizing knowledge transfer. 2. Aligning dimensions by mapping functions. Before knowledge transfer, we will first map the knowledge vectors of the current domain to the feature space of the target domain through MLP. The embedding dimensions of the source and target domains can be aligned by changing the input dimension of the MLP.

---

### Author Rebuttal · Authors · 2024-08-06

We sincerely thank all the reviewers for their valuable comments and suggestions, which are crucial for improving our work. Here we carefully response your questions point-by-point, and more details of the responses are **in the PDF file** at the bottom of this **Author Rebuttal.**

**1:** How well the model generalizes to other datasets or real-world scenarios with different characteristics.

**Response:** As per your requests, we conducted new experiments on the Douban dataset to validate the generalizability of our approach. The dataset information (Table u1) and experimental results (Table u2) are available in the rebuttal PDF file.

- *Experimental settings:* We filtered users and items in the dataset with a number of interactions less than 5. Since the Douban dataset contains only three domains, we set them as target domains for the experiments in turn.
- *Results:*  As shown in Table u2, first, our method outperforms all the baselines in almost all metrics, except NDCG@5 on Movie domain. For the only exception to the NDCG@5 metric, our method is the second best. Although **Single domain** is better than our methods on NDCG@5, it has poor performance on the other three metrics (HR@5, HR@10, NDCG@10) because it can not leverage knowledge form source domains. Second, for the CDR scenario, our method demonstrated its superiority compared to other CDR methods. Third, the inclusion of Gaussian noise for the purpose of preserving inter-domain privacy does not always have a side effect on model performance (e.g., in the book domain,  FedGCDR-DP outperforms FedGCDR on the metrics HR@5, NDCG@5 and NDCG@10), and we believe that the inclusion of noise benefits the model's attention mechanism enhancing its ability to filter out noise and negative knowledge. In conclusion, the experimental results demonstrate that FedGCDR significantly outperforms state-of-the-art methods.

---

### Decision · Program_Chairs · 2024-09-25

**Decision:**

Accept (poster)

**Comment:**

This paper proposes FedGCDR, a novel federated graph learning framework for cross-domain recommendation that effectively addresses privacy concerns and the risk of negative transfer. The framework features a positive knowledge transfer module that leverages differential privacy to securely extract reliable domain knowledge, and a knowledge activation module that filters out harmful knowledge to enhance target domain training. Extensive experiments on the Amazon dataset demonstrate that FedGCDR significantly outperforms state-of-the-art methods. Reviewers agree that the problem studied is meaningful and the proposed method has clear merits. We appreciate the authors' efforts in the rebuttal to address reviewers' comments and provide additional experimental results. I recommend acceptance, with the suggestion that the authors carefully address the additional feedback and suggestions raised by reviewers if the paper is accepted at NeurIPS.